# Molecular Capture of *Mycobacterium tuberculosis* Genomes Directly from Clinical Samples: A Potential Backup Approach for Epidemiological and Drug Susceptibility Inferences

**DOI:** 10.3390/ijms24032912

**Published:** 2023-02-02

**Authors:** Rita Macedo, Joana Isidro, Rita Ferreira, Miguel Pinto, Vítor Borges, Sílvia Duarte, Luís Vieira, João Paulo Gomes

**Affiliations:** 1National Reference Laboratory for Mycobacteria, Department of Infectious Diseases, National Institute of Health (INSA), 1649-016 Lisbon, Portugal; 2Genomics and Bioinformatics Unit, Department of Infectious Diseases, National Institute of Health (INSA), 1649-016 Lisbon, Portugal; 3Innovation and Technology Unit, National Institute of Health (INSA), 1649-016 Lisbon, Portugal

**Keywords:** *Mycobacterium tuberculosis*, RNA-baits, molecular capture, target enrichment, whole genome sequencing, resistance, surveillance

## Abstract

The application of whole genome sequencing of *Mycobacterium tuberculosis* directly on clinical samples has been investigated as a means to avoid the time-consuming need for culture isolation that can lead to a potential prolonged suboptimal antibiotic treatment. We aimed to provide a proof-of-concept regarding the application of the molecular capture of *M. tuberculosis* genomes directly from positive sputum samples as an approach for epidemiological and drug susceptibility predictions. Smear-positive sputum samples (*n* = 100) were subjected to the SureSelectXT HS Target Enrichment protocol (Agilent Technologies, Santa Clara, CA, USA) and whole-genome sequencing analysis. A higher number of reads on target were obtained for higher smear grades samples (i.e., 3+ followed by 2+). Moreover, 37 out of 100 samples showed ≥90% of the reference genome covered with at least 10-fold depth of coverage (27, 9, and 1 samples were 3+, 2+, and 1+, respectively). Regarding drug-resistance/susceptibility prediction, for 42 samples, ≥90% of the >9000 hits that are surveyed by TB-profiler were detected. Our results demonstrated that *M. tuberculosis* genome capture and sequencing directly from clinical samples constitute a potential valid backup approach for phylogenetic inferences and resistance prediction, essentially in settings when culture is not routinely performed or for samples that fail to grow.

## 1. Introduction

Tuberculosis (TB) remains one of the most important infectious diseases globally [1]. The gold standard for the routine clinical diagnosis and drug susceptibility testing (DST) for *Mycobacterium tuberculosis* is culture-based, which requires months for visible growth, leading to a potential prolonged suboptimal antibiotic treatment [2]. Establishing a resistance profile from the initial TB diagnosis is a priority. Indeed, although there are several molecular assays already endorsed by the World Health Organization (WHO), they fall short on targets and resistance-related regions or genes, to assure a correct prediction of resistance [3]. The potential of Whole Genome Sequencing (WGS) as a diagnostic assay has been repeatedly demonstrated and enables a comprehensive identification of all known resistant mutations for all TB drugs as well as it can provide reliable contact tracing information [4,5,6,7,8,9]. For these reasons, WGS-based methodologies have been implemented as a routine for early positive cultures identification, resistance prediction, and surveillance at the National Reference Tuberculosis Laboratory (NRL-TB) of the Portuguese National Institute of Health (NIH) [10,11]. This approach performs at a comparable cost to phenotypic assays offering short turnaround times. However, even early positive cultures may represent some weeks of bacterial growth. In this context, generating WGS information directly from samples (bypassing the time-consuming culture) would constitute a tremendous achievement towards a rapid DST-informed diagnosis. It has, however, a major potential hurdle as biological samples contain variable amounts of human cells (mixed with *M. tuberculosis* cells) that can account for up to 99.9% of the total DNA [12,13,14], which results in high human to bacterial DNA content ratio with consequent low depth of coverage and poor identification of resistance [13]. One of the alternatives proposed, and already in use for routine diagnostic purposes, consists in sequencing only regions associated with drug resistance (DR). Targeted sequencing (TS) panels have shown effectiveness in obtaining resistance profiles directly from clinical samples, providing a signature of genetic markers associated with drug resistance, and promoting a personalized treatment [15,16]. It also increases the sequencing depth, facilitating the identification of subpopulations of susceptible and resistant bacteria (heteroresistance), which can impact the early diagnosis of DR-TB [17]. However, a TS approach does not provide information regarding the identification of novel resistant markers and does not allow genomic epidemiology and transmission inferences. As such, the ability to directly sequence the complete genome of *M. tuberculosis* from clinical specimens of infected patients would be the logical step forward to deliver the full potential of WGS for TB control.

Several studies in multiple areas of infectious diseases have already described the use of specific protocols relying on custom designed RNA oligonucleotides spanning the entire microbial genome, which can recover by hybridization (i.e., “target enrichment”) low copy numbers of DNA directly from clinical samples with sufficiently high sensitivity and specificity to enable efficient WGS [18,19,20,21]. However, mycobacterial cells may aggregate because of the high mucus content of respiratory samples, meaning that the volume and Acid-Fast Bacilli (AFB) count may not represent the total quantity of mycobacteria available [12,13,21]. Therefore, clinical samples require pre-processing for homogenization and enrichment purposes and depletion of non-*Mycobacterium* cells/DNA.

The main objective of this study is to provide a proof-of-concept regarding the application of the molecular capture of *M. tuberculosis* genome sequences directly from positive sputum samples collected from TB patients as a potential backup approach for epidemiological and drug susceptibility inferences.

## 2. Results

### 2.1. Impact of Sample Characteristics (Smear Grade and Human/Bacteria Load) on Genome Capture Success

For the sake of clarity, all data regarding samples’ characterization and WGS-associated data are summarized in Appendix A. All 100 smear-positive sputum samples were subjected to the SureSelect^XT HS^ Target Enrichment protocol (Agilent Technologies, Santa Clara, CA, USA; see Methods). Among these, 48 were 3+ smear samples, 33 were 2+ and 19 were 1+. For 30 samples, we ended up with an amount of DNA range of 2.4–9.9 ng, which is slightly lower than the recommended minimum of 10 ng as DNA input.

The number of *M. tuberculosis* (mean = 3821.5 *katG* copies) and human (mean = 146,626.5 *β-actin* copies) cells per µL were determined for 81 samples. For the remaining 19 samples, the total volume of the extracted DNA was used in the target enrichment protocol. While higher numbers of human cells were expectedly associated with higher DNA inputs, the number of *M. tuberculosis* was higher for lower DNA inputs (Appendix A). Moreover, comparing the three smear categories, lower smear grades correlated with lower *M. tuberculosis* loads and higher number of human cells, while 3+ samples showed the highest *M. tuberculosis* loads and lowest number of human cells (Figure 1).

Most samples (64/100) generated a final library with a molarity below 0.5 nM (minimum recommended for loading the sequencing apparatus Illumina NextSeq 550), leading to the need of concentrating the pooled libraries in a SpeedVac system. Of note, this was not due to low initial DNA input since the two variables were inversely proportional to each other (Appendix A). Even with the addition of a concentrating step, that allowed a better normalization of libraries and an increase in the flow cells output, the generated number of reads per sample was highly variable and ranged between 306 and 30,635,472 paired-end reads (mean = 6,170,816.36; median = 2,655,720) (Figure 1C).

As a means of assessing the success of the target enrichment, we determined the percentage of reads “on target” (reads that mapped against the *M. tuberculosis* H37Rv reference genome) and the percentage of reads classified by Kraken2 as *Mycobacteriaceae* family (or lower taxonomic levels). Expectedly, these two metrics were highly correlated and directly proportional (Appendix A). Overall, we obtained promising results of target enrichment for a high number of samples, with 78/100 samples having at least 50% of reads “on target”, and 44 of these having more than 90% of reads “on target”. Moreover, the number of raw reads and reads on target was higher for higher smear grades samples (Figure 1 and Figure 2).

However, the success in the capture of *M. tuberculosis* from biological samples does not indicate that enough reads were obtained for a given sample to be suitable for downstream analysis (i.e., requiring enough horizontal and vertical genome coverage). For example, only for 37 out of 100 samples we obtained showed ≥90% of the reference genome covered with at least 10-fold depth of coverage (see methods for details), which we routinely use in our laboratory as minimum requirements for genome-based surveillance of *M. tuberculosis*. Of these, 27, 9, and 1 were 3+, 2+, and 1+ smear grade samples, respectively, showing that success can be obtained for samples with low *M. tuberculosis* content (Figure 2).

### 2.2. SNP-Based Core Genome Analysis

Aiming at understanding the usefulness of the *M. tuberculosis* genomes captured through the target enrichment approach for genomic surveillance purposes (e.g., to study phylogenetic relationships), we analysed the 37 samples for which we obtained ≥90% of the reference genome covered with at least 10-fold depth of coverage, following a single nucleotide polymorphism (SNP)-based core genome analysis (Appendix A). The generated minimum spanning tree was based on 3441 variable positions and allowed the detection of four genetic clusters. In four patients, each with two samples available for analysis, no SNP differences were found among the pairs of DNA samples, reinforcing the reproducibility and robustness of the methodology.

### 2.3. Drug-Resistance/Susceptibility Prediction

All samples were subjected to drug-resistance screening directly from reads using TB-profiler [22,23]. Since some of the genomic regions screened have homology with other bacterial species (e.g., *rrs* gene), the raw data were firstly filtered to exclude non-Mycobacterium reads and reduce false-positive hits in these regions. With this approach, 68 and 42 samples could be screened in ≥50% and ≥90% of the >9000 nucleotide positions (“hits”) analysed by TB-profiler, respectively. Among the 42 samples for which ≥90% hits were covered, 24 had 100% hits covered. Moreover, it was possible to identify a multidrug resistant strain and confirm the fully susceptible profile associated with most samples (Appendix A). As 18/42 samples lacked coverage in some positions, mostly associated with the *rrs* gene, we repeated the same analysis by reducing by half the depth of coverage required to validate a position. This allowed the validation of 100% of the positions in 32 samples, without affecting the mutations identified previously with the default cut-off depth of coverage of 10 reads.

Importantly, this analysis also allowed the identification of mutations in minor populations in 6/42 samples, or 12/100 when considering lower coverage samples (with proportions between 11% and 25% associated with resistance to isoniazid, pyrazinamide, ethambutol, and aminoglycosides; Appendix A), that could have been missed if sequencing had been performed from a cultured strain and not directly from the clinical sample.

## 3. Discussion

In the present study, we provide a proof-of-concept on the value of molecular capture of *M. tuberculosis* genome sequences directly from positive sputum samples for resistance prediction, genomic surveillance, and potential evaluation of intra-host genetic diversity. The capture of large genomic sequences directly from clinical samples, such as the baits-based Agilent SureSelect procedure, has been widely used in genetic-related diagnoses such as hemoglobinopathy cases [24] and viruses typing [25]. More recently, its role in the surveillance and diagnostic of *M. tuberculosis* has been exploited [12,13,14,20,21] with uncertain efficiency and with results not easily comparable due to different methodologies and/or sample selection criteria. For instance, Nimmo et al. (2019) followed a similar enrichment protocol (SureSelect) but no information is available regarding AFB counts or *M. tuberculosis* quantification of the samples processed, which hampers the comparison of success rates [26]. Similarly, Doyle et al. (2018) used the SureSelect XT approach and reported high success rates, even among AFB 1+/scanty samples [27]. Nonetheless, in line with our results, the authors also observe the best results with higher smear grades, showing that AFB counts can be a good predictor of success [27]. Despite launching important clues about the usefulness of this approach in *M. tuberculosis*, these studies mostly focused on the drug susceptibility issue and used limited sample datasets. In our study, we provide a detailed characterization of 100 samples, not only regarding the analytic parameters but also the upstream sample traits that can ultimately affect the outcome of the protocol. Furthermore, we tried to mimic a real diagnostic scenario where each sample was sequenced only once. For instance, some samples with a good percentage of reads “on target” but low number of reads could have been recovered with new rounds of sequencing to increase genome coverage.

*M. tuberculosis* is particularly appropriate for the use of diagnostic WGS with enrichment, since unlike the majority of pathogenic organisms, it has a well-characterized clonal nature, with low levels of sequence variation, and does not undergo recombination or horizontal transfer [28]; thus, a stable set of oligonucleotide baits can be created/designed and sequencing data can be mapped against a reference genome. By overcoming the constraints of time-consuming laboratory procedures for the isolation of *M. tuberculosis* strains in culture, we were able to provide a methodology that allowed not only the identification and prediction of genotypic resistance, but also the possibility to integrate this approach in a “real-time” surveillance for the rapid articulation with the public health authorities. Within a 5-day wet-lab procedure, after DNA isolation directly from sputum samples, we can retrieve all the information needed for routine diagnostic purposes, skipping the 1–3 weeks period required for culture isolation. Furthermore, a rough estimation on the total cost of this methodology showed, comparing with WGS-based analysis from strains isolated in culture, an increase in about EUR 120–150. Thus, an estimated final cost would roughly be below EUR 200 per sample, depending on the desirable coverage and on the sequencing equipment and flow cell that is used. However, it is important to note that other implementations that vary in both turnaround times and final costs are possible.

The decision regarding which samples should undergo additional enrichment with RNA baits and the robustness of the computational analysis is crucial for the success of the direct-WGS workflow. As expected, samples with higher AFB counts and consequently a higher number of *katG* copies (indirect estimate of the number of *M. tuberculosis* cells) were more likely to provide confident results. For example, we obtained a higher number of reads on target for higher smear grades samples (i.e., 3+ followed by 2+) (Figure 1). Contrarily, the number of *β-actin* copies seems to be inversely correlated with the number of *katG* copies and, hence, the success of the procedure. Although in a speculative basis, we believe this can be related with the sample collection, as a patient with a higher *M. tuberculosis* load may yield samples enriched in mucous/bacteria rather than in human cells. This is also illustrated by the results obtained among the 37 out of 100 samples showing ≥90% of the reference genome covered with at least 10-fold depth of coverage, as 27, 9, and 1 samples were 3+, 2+, and 1+ smear grade samples, respectively, (Figure 2). However, these data also show that samples with lower AFB/*katG* counts could also yield good results, as the bioinformatics WGS data analysis improvements (including removal of non-mycobacteria reads and establishment of “success” thresholds for analysis) allowed the successful inclusion of several samples with smear results <3+. The drug-resistance/susceptibility prediction was not an exception as the 42 samples for which ≥90% of the >9000 hits surveyed by TB-profiler were covered, included several 2+ samples.

Another relevant application of direct WGS is the study of *M. tuberculosis* genetic diversity in sputum samples, which might better reflect the within-patient bacterial populations. Unlike WGS performed from DNA isolated from pure cultures, which potentially leads to the loss of information on the existence of sub-populations, sequencing directly from clinical samples can provide information on the real scenario of the in vivo sub-populations that might co-exist during the infection period. Once more, this can be illustrated for 12 out of the 100 samples as we identified mutations in minor populations (ranging from 11% to 25% intra-patient frequency) associated with resistance to several antibiotics.

There are limitations to surpass before direct WGS approaches can be used to support the control of the TB pandemic. All these workflows are unaffordable in the majority of high/medium TB-burden countries, and sensitivity remains low compared with culturing. However, in terms of drug resistance and epidemiological surveillance, genome capture coupled with WGS enables the study of TB transmission dynamics and resistance in countries where culture and drug susceptibility testing are not routinely performed. It also constitutes a potential valid backup approach for non-viable samples. In this regard, it would have particular interest for samples predicted (by rapid molecular tests) to contain multi-drug resistance strains, for which it would be important to determine not only a more complete set of resistant hits, but also the phylogenetic context.

## 4. Materials and Methods

### 4.1. Samples Description

Smear-positive sputum samples (*n* = 100) retrieved from pulmonary TB patients and that were sent to the NRL-TB from the Portuguese NIH for routine diagnostic purposes were tested (Appendix A). Samples were decontaminated using N-acetyl-l-cysteine/NaOH (1% NaOH final concentration) and resuspended after centrifugation in 2 mL phosphate buffer (pH 6.8). After inoculation for phenotypic testing, all the remaining sputum specimens were kept frozen at −20 °C until further use. The set was composed of samples with different positive AFB scoring of 1+, 2+, and 3+ (visually quantified according to WHO guidelines) in order to test the target enrichment protocol with different *M. tuberculosis* smear grades. 

### 4.2. Phenotypic Resistance Profiles

All isolates were phenotypically tested for susceptibility to first-line drugs rifampicin (RIF), isoniazid (INH), ethambutol (EMB), pyrazinamide (PZA), and streptomycin (STR). Isolates resistant to at least RIF and INH (i.e., representing multidrug-resistant TB [MDR-TB]) were additionally tested for susceptibility to kanamycin (KAN), amikacin (AMK), ofloxacin (OFL), capreomycin (CAP), ethionamide (ETH), prothionamide (PTH), and para-aminosalicylate sodium (PAS). Drug susceptibility testing (DST) was carried out on an automated liquid medium-based system, Bactec MGIT960 (Becton Dickinson, East Rutherford, NJ, USA), using standard drug concentrations (in micrograms per milliliter) as follows: for STR: 1.0; for INH: 0.1; for RIF: 1.0; for EMB: 5.0; for PZA: 100.0; for OFL: 2.0; for AMK: 1.0; for CAP: 2.5; for KAN: 5.0; for ETH: 5.0; for PTH: 2.5; and for PAS: 4.0.

### 4.3. DNA Extraction

After heat killing of the bacteria (95 °C for 1 h), high-quality DNA samples were prepared using the QIAamp DNA Mini Kit (Qiagen, Düsseldorf, Germany) according to the manufacturer’s protocol. 

### 4.4. Generation of Standard Curves for Real-Time Quantitative PCR (qPCR)

To quantify the number of *M. tuberculosis* genomes in each sample, a plasmid standard curve was generated as previously described for other pathogens [19,29,30]. Primers for the conserved *M. tuberculosis* single-copy gene *katG* were designed based on constant regions (primers KatG-A TTACCGCTGGGCGTGTTC and KatG-B TCACGAAGAAGTCGTTGGTCAGT by using Primer Express software v3.0; Applied Biosystems, Waltham, MA, USA), according to the sequence of MTB H37Rv strain (Genbank # AL123456). An amplified fragment (58 bp) of *katG* was cloned into the pCR^®^ 2.1 vector using the TOPO TA technology (Invitrogen, Waltham, MA, USA) according to the manufacturer’s instructions. After transformation of DH5α *E. coli* with the cloned vector and subsequent overnight propagation, plasmid DNA was purified and transformation was confirmed by PCR and sequencing. The plasmid copy number was calculated according to the formula: Nº plasmid/mL = (Avogadro’s No. × Plasmid conc. (g/mL))/MW of 1 mol of plasmids (g). The standard curve consisted of eight serial plasmid dilutions (~1 to 1 × 10^8^ plasmid copies/μL). The number of human cells/sample was quantified by a similarly generated plasmid standard curve using an amplified fragment (73 bp) of a single copy human gene (*β-actin*) cloned in a similar vector, according to Gomes et al. 2006 [29] (primers β-actin-3 GGTGCATCTCTGCCTTACAGATC and β-actin-4 ACAGCCTGGATAGCAACGTACAT).

### 4.5. qPCR for Quantification of MTB vs. Human Cells

The real-time quantification was performed using the Light-Cycler^®^ 480 SYBR Green chemistry and optical plates (Roche Diagnostics, Basel, Switzerland). The qPCR reagents consisted of 2 × SYBR Green I Master Mix, 400 nM of each primer, and 5 µL of DNA sample in a final volume of 25 µL. The thermocycling profile was: 10 min/95 °C followed by 40 cycles of 15 s/95 °C and 1 min/60 °C. Specificity was checked by generating the dissociation melting curves. Absolute quantification of bacterial and human genomes was calculated in relation to the respective plasmid standard curve. The relative load of *M. tuberculosis* cells in each sample was determined as the ratio between the number of *katG* and *β-actin* copies.

### 4.6. DNA Capture Directly from Clinical Samples: SureSelect^XT HS^ Target Enrichment

In order to capture the *M. tuberculosis* DNA directly from clinical samples, complementary RNA oligonucleotide “baits”, 120 bp in size, were designed to span the ~4.5 Mb of the *M. tuberculosis* genome. As such, the reference genome sequence of the MTB H37Rv strain (Genbank #AL123456) was in silico fragmented into 120 bp sequences twice, to ensure an overlap of 60 bp between sequences. Due to their rich GC content, which can interfere with DNA capture, all *M. tuberculosis* genes of the PE, PPE, and PE-PGRS family were also independently fragmented into 120 bp sequences in order to increase capture sensitivity. All resulting sequences were BLASTn searched against the Human Genomic + Transcript database to excluded homologous sequences to the human genome. Overall, a total of 42,278 RNA probes were generated and this custom bait library was then uploaded to the SureDesign software (https://earray.chem.agilent.com/suredesign, accessed on 4 August 2016) and synthesized by Agilent Technologies (Santa Clara, CA, USA). During synthesis, the 2198 sequences complementary to the PE, PPE, and PE-PGRS family were unbalanced 8:1 to potentiate capture.

For the libraries preparation, the SureSelect^XT HS^ Target Enrichment System for Illumina Paired-End Multiplexed Sequencing Library (Agilent Technologies, Santa Clara, CA, USA) procedure was used (version E1, April 2021). The “Preparation of high-quality gDNA from fresh biological samples” instructions were followed in Step 1. to ensure high quality gDNA yield (see DNA extraction in section above). In order to calibrate the input to 10–200 ng in 7 µL, the gDNA samples were quantified using Qubit^HS^ kit (Invitrogen, Life Technologies, USA), and subsequently fragmented using the Agilent’s SureSelect^XT HS^ Low input Enzymatic Fragmentation kit (“Method 2: Enzymatic DNA Fragmentation” option described in Step 2), according to manufacturer’s instructions. Library preparation was then resumed at Step 3, “Repair and dA-Tail the DNA Ends” and carried out with no further alterations. Distribution of library fragments was evaluated using the Fragment Analyzer (Agilent, Santa Clara, CA, USA) and the PROSize 3.0 analysis software. Library concentration was obtained following smear analysis of fragment sizes. Pools of 16-indexed libraries were sequenced in the Illumina NextSeq 550 instrument (Illumina, San Diego, CA, USA) according to the manufacturer’s instructions and the sequencing run recommendations of the SureSelect^XT HS^ Target Enrichment System protocol (Agilent Technologies, Santa Clara, CA, USA).

### 4.7. Sequence Data Analysis

Raw reads were subjected to quality analysis and trimming using the INNUca v4.2.2 pipeline [31] (which integrates FastQC v0.11.5 (http://www.bioinformatics.babraham.ac.uk/projects/fastqc/) and Trimmomatic v0.36) [32]. To access the percentage of reads on target, the improved and trimmed reads were mapped using Bowtie 2 v2.4.2 against the reference MTB H37Rv strain genome (Genbank #AL123456). In parallel, Kraken2 v2.0.7-beta (https://github.com/DerrickWood/kraken2) [33] was also used to determine the percentage of reads classified as *Mycobacteriaceae* family (or lower taxonomic levels).

#### 4.7.1. In Silico Drug Resistance Prediction

For in silico drug resistance prediction and to minimize the issues related to genes with homology in different bacterial species (such as *rrs*), the trimmed reads were filtered using the seqtk v1.3-r106 tool (https://github.com/lh3/seqtk) and only the reads classified as *Mycobacteriaceae* family (or at a lower taxonomic level) by Kraken2 were kept. The filtered reads were then analysed using TB-profiler v4.2.0 with minimum depths of coverage of 10 and 5 for comparison.

#### 4.7.2. SNP-Based Core-Genome Analysis

For the SNP-based analysis, the trimmed reads from all samples were mapped against the MTB H37Rv strain genome using Snippy v4.5.1 (https://github.com/tseemann/snippy) and the variant positions were validated with a minimum depth of coverage of 10-fold, minimum proportion of 70% of reads showing the alternative allele, a minimum mapping quality of 30, and a minimum base quality of 20. The core single nucleotide variants (SNVs) were extracted using the Snippy core module in Snippy, and only the samples with at least 90% of the reference genome covered and at least 10-fold depth of coverage were validated for the final core-SNV phylogeny. To minimize bias in the phylogeny, core-SNV falling within known *M. tuberculosis* genomic regions with high GC content or repetitive elements, as well as known SNVs in resistance-associated positions, were excluded (compiled by Kohl and colleagues, available at https://github.com/ngs-fzb/MTBseq_source/tree/master/var/res, accessed on 31 July 2018) using the “mask” parameter of Snippy core. A core-SNV minimum spanning tree of the validated samples was generated using Grapetree v1.5.0 (https://github.com/achtman-lab/GrapeTree) [34].

## Figures and Tables

**Figure 1 ijms-24-02912-f001:**
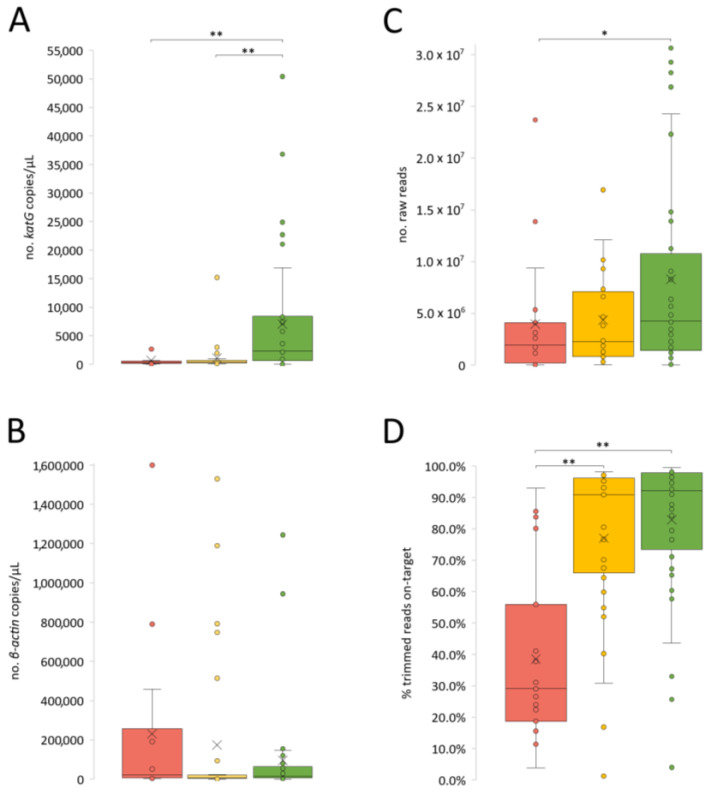
Characterization of the studied smear-positive samples (per smear grade), according to the: (**A**) number of *katG* copies/μL (as an indirect estimate of the number of *M. tuberculosis*/μL), (**B**) *β-actin* copies/μL (as an indirect estimate of the number of human cells/μL), and (**C**) No. of raw reads and (**D**) % trimmed reads on target (more details in Appendix A). Differences between smear groups were tested using the Mann–Whitney U test. * *p*-value < 0.05; ** *p*-value < 0.01. All considered metrics are higher for higher smear grades (2+ and 3+) with the exception of *β-actin* copies, which is higher for the 1+ samples, although not statistically significant.

**Figure 2 ijms-24-02912-f002:**
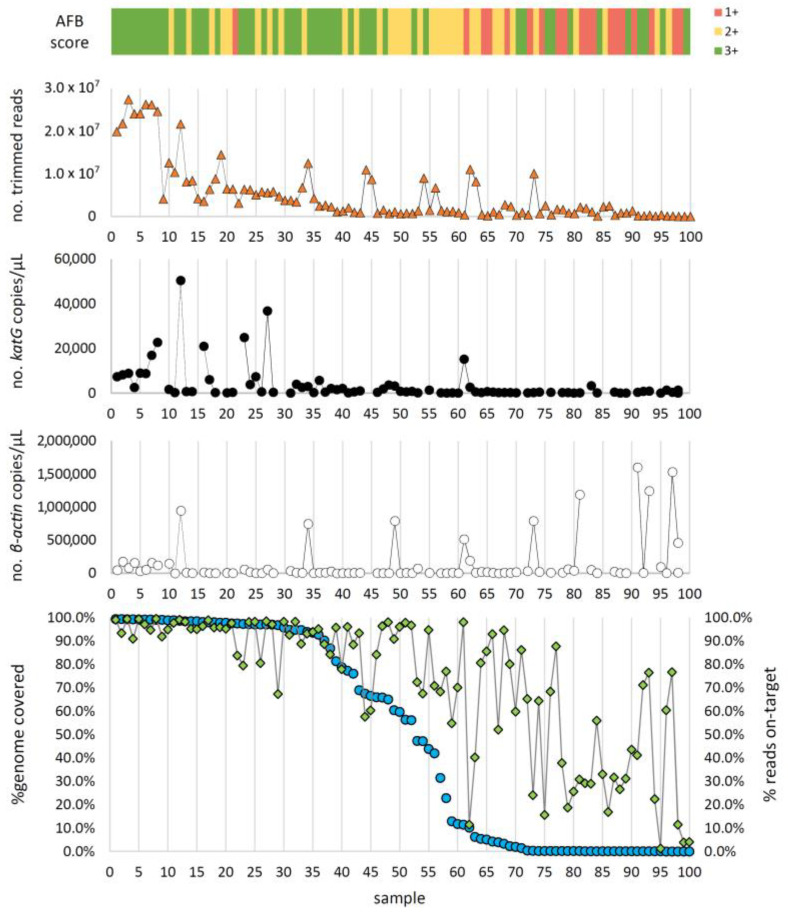
Overview of the relation between genome coverage (bottom panel) and multiple factors (microscopy grade, No. of trimmed reads, No. of *katG* copies and No. of *β-actin* copies), ordered from the sample with the highest to the lowest % genome coverage. Samples with higher genome coverages correlate with higher smear grades (top panel), number of *katG* copies and, expectedly, number of reads generated. *β-actin* copies show the opposite pattern as lower number of copies correlate with higher genome coverages. As for the percentage of reads “on target”, it is highly variable as, for instance, a high genome coverage also depends on the number of reads, even if the capture/enrichment is successful.

## Data Availability

Sequence data (only reads mapping against *M. tuberculosis* H37Rv reference genome) generated in this study were deposited in the European Nucleotide Archive under the Bioproject PRJEB59106. The set of RNA baits sequences used in the current Target Enrichment protocol are available at https://doi.org/10.5281/zenodo.7550025.

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
