# Peer review of "Molecular Capture of Mycobacterium tuberculosis Genomes Directly from Clinical Samples: A Potential Backup Approach for Epidemiological and Drug Susceptibility Inferences"

_ijms, 2023, doi:10.3390/ijms24032912_

Round 1

Reviewer 1 Report

The manuscript is meticulously written, I have attached some minor comments.

Author Response

Dear Editor,

We thank you and the reviewers for the opportunity to resubmit the current study. Please find below our answers to the reviewers’ concerns, which we believe we have completely addressed. We have also improved the quality of the figures, added the p-values to Figure 1 and included the accession numbers in Table S1.

Reviewer #1

The manuscript is meticulously written, I have attached some minor comments.

The manuscript focuses to provide proof-of-concept regarding the application of the molecular capture of M. tuberculosis genome sequences directly from positive sputum samples collected from TB patients as the potential backup approach for epidemiological and drug susceptibility inference. The manuscript is written nicely, data were analyzed carefully and results were presented meticulously. I, however, have some

  1. In Figure 1, the author should provide p-values for comparison between diferente grades of the smear-positive category. Also, please convey short results in the figure legends, and apply this to other figures and tables too.

Response: We have now added the p-values of the comparisons between smear groups to Figure 1 and have updated the figures legends.

  1. For Figure 1(B), the author should explain the reason for the lower numbers (reads of) β-actin copies/µL in the figure legends to better understand the readers.

Response: We thank the reviewer for the comment. We have added a sentence in discussion about this topic.

  1. There was no “scanty” category in the smear grades. The author should also explain the reason for that. It would be nice to see the role of WGS in the scanty category when the specimen is on a very low bacterial load.

Response: We agree with the reviewer that the “scanty” category could be interesting to include. However, as our results showed that lower AFB counts yield unequivocally worse results,  we have decided not to further include “scanty” samples. We believe that the inclusion of the three categories (1+, 2+ and 3+) enrol the vast majority of the samples of the “real world” and the inclusion of  the “scanty category” would not change the take-home message as the most useful data was obtained with 2+ and 3+ samples.

  1. To improve readability authors should provide a detailed flowchart, explaining the collection specimens, processing, and result generation. This will help follow the train of thought for what is explained in the text.

Response: Done. The flowchart is now our Figure S2.

  1. There are a lot of small acronyms and short-form used, if the journal permits, the author should write a small abbreviation sub-section explaining all small details.

Response: We carefully checked the whole manuscript and now all acronyms/abbreviations are described when they first appear in the text. We believe the IJMS guidelines do not allow a separate list of acronyms/abbreviations in the manuscript.

Reviewer 2 Report

Overall, the whole structure of this study is good and some corrections are recommended for providing clear information. Particularly, I listed the following comments in detail here.
Major concerns:

In abstract, the author needs to mention the ingredients of methods, and materials. Also, the finding of the assay could be added step by step based on material and method. I recommend considering regularly assays and results. Besides, name of bacteria should be italic. All of the names and terms should be completely mentioned for the first time in text and so on. For example, WGS.

In introduction, some sentences lack reference. For example, “This approach performs at a comparable cost to phenotypic assays offering short turnaround times. However, even early positive cultures may still represent some weeks of bacterial growth. In this context, generating WGS information directly from samples (bypassing the time-consuming culture) would constitute a tremendous achievement towards a rapid DST-informed diagnosis.”, “However, a TS approach will not provide information regarding the identification of novel resistant markers and will not allow genomic epidemiology and transmission inferences. As such, the ability to directly sequence the complete genome of M. tuberculosis from clinical specimens of infected patients would be the logical step forward to deliver the full potential of WGS for TB control.”, “Current evidence suggests that cancer immune cell populations are not randomly distributed but follow a specific pattern”. All of the names and terms should be completely mentioned for the first time such as WHO.

In methods, the author needs to mention the ingredients of methods, locations, and materials. Please add references for all tests.

In the discussion, discuss your results before relating them to the results of other published work. Also, the author must step by step to come to the results and comparison with others.

What is your conclusion? Do the authors have more thoughts on this field?

Author Response

Dear Editor,

We thank you and the reviewers for the opportunity to resubmit the current study. Please find below our answers to the reviewers’ concerns, which we believe we have completely addressed. We have also improved the quality of the figures, added the p-values to Figure 1 and included the accession numbers in Table S1.

Reviewer #2

Overall, the whole structure of this study is good and some corrections are recommended for providing clear information. Particularly, I listed the following comments in detail here.

Major concerns:

In abstract, the author needs to mention the ingredients of methods, and materials. Also, the finding of the assay could be added step by step based on material and method. I recommend considering regularly assays and results. Besides, name of bacteria should be italic. All of the names and terms should be completely mentioned for the first time in text and so on. For example, WGS.

Response: Multiple modifications were done throughout the manuscript to meet the reviewer’s comments.

In introduction, some sentences lack reference. For example, “This approach performs at a comparable cost to phenotypic assays offering short turnaround times. However, even early positive cultures may still represent some weeks of bacterial growth. In this context, generating WGS information directly from samples (bypassing the time-consuming culture) would constitute a tremendous achievement towards a rapid DST-informed diagnosis.”,

Response: These sentences refer to the same references cited in the sentence immediately before (Refs 10 and 11). As we start the sentence with “This approach”, we believe that it is clear that it is a continuation of the previous sentence so we would prefer not to repeat the references in order to avoid redundancies.

 “However, a TS approach will not provide information regarding the identification of novel resistant markers and will not allow genomic epidemiology and transmission inferences. As such, the ability to directly sequence the complete genome of M. tuberculosis from clinical specimens of infected patients would be the logical step forward to deliver the full potential of WGS for TB control.”

Response: We believe this sentence is self-explanatory since it refers to “Targeted sequencing” (TS), in which only small sequences of interest (AMR-associated genes) are sequenced.

 “Current evidence suggests that cancer immune cell populations are not randomly distributed but follow a specific pattern”.

Response: This sentence is not present in the submitted manuscript.

All of the names and terms should be completely mentioned for the first time such as WHO.

Response: Done.

In methods, the author needs to mention the ingredients of methods, locations, and materials. Please add references for all tests.

Response: Done.

In the discussion, discuss your results before relating them to the results of other published work. Also, the author must step by step to come to the results and comparison with others.

Response: We have now adjusted the discussion to include a brief description of our results in the beginning of the discussion and have further compared our results with other studies (see discussion section). Regarding the general structure of the discussion, we would like to keep it as it is as we first present the “problem” to overcome and then describe and discuss our results. .

What is your conclusion? Do the authors have more thoughts on this field?

Response: The major conclusions are stated in the last two paragraphs: “…relevant application of direct WGS is the study of M. tuberculosis genetic diversity in sputum samples, which might better reflect the within-patient bacterial populations”, …, “genome capture coupled with WGS enable the study of TB transmission dynamics and resistance in countries where culture and drug susceptibility testing are not routinely performed”, …” It also constitutes a potential valid backup approach for non-viable samples”, …, “it would have particular interest for samples predicted (by rapid molecular tests) to contain multi-drug resistance strains, for which it would be important to determine not only a more complete set of resistant hits, but also the phylogenetic context”.

Reviewer 3 Report

The authors studied the M. tuberculosis genome capture and sequencing directly from clinical samples such as potential valid backup approach for phylogenetic inferences and resistance prediction.

I found some published manuscripts with the same scope. Authors should clarify what the contribution of their research is. Must be improve the introduction.

With the data in Figure 1, I suggest performing statistics to compare the groups.

I suggest a table that displayed the comparison about resistance obtained with phenotype and genotype.

I think this paper don’t have the scope of special issue CRISPR-Cas in Genomic Manipulation and Antimicrobial Resistance

I detected some mistakes:

1. sometimes M. tuberculosis is not in italic style.

2. line 100 and 101 missing μ

3. línea 144 Figura complementaria 1. debe ser Figura 3.

4. All the references must be checked; they are not in accordance with the style of journal.

Author Response

Dear Editor,

We thank you and the reviewers for the opportunity to resubmit the current study. Please find below our answers to the reviewers’ concerns, which we believe we have completely addressed. We have also improved the quality of the figures, added the p-values to Figure 1 and included the accession numbers in Table S1.

Reviewer #3

The authors studied the M. tuberculosis genome capture and sequencing directly from clinical samples such as potential valid backup approach for phylogenetic inferences and resistance prediction.

I found some published manuscripts with the same scope. Authors should clarify what the contribution of their research is. Must be improve the introduction.

Response: We understand the reviewer’s comment. However, as the first large paragraph of the discussion focus precisely on the outputs of our study when compared with previous works, the inclusion of this topic in the introduction would create a redundancy that we would like to avoid. Instead, we opted to include in the introduction the need to apply this kind of approaches for epidemiological and drug susceptibility inferences. Moreover, we further discuss our work’s contribution and how it compares to previous studies (see discussion section).

With the data in Figure 1, I suggest performing statistics to compare the groups.

Response: We acknowledge the reviewer for this important input.  We have now added the p-values of the comparisons between smear groups to Figure 1.

I suggest a table that displayed the comparison about resistance obtained with phenotype and genotype.

Response: Since 19 samples were not tested for drug resistance and that, among the 81 samples with available phenotypes, 73 were fully susceptible, we believe that this table would not be informative. For this reason, we instead determined the TB-profiler positions that were covered and describe in the text that the only multidrug resistant strain in the dataset could be fully characterized.

I think this paper don’t have the scope of special issue CRISPR-Cas in Genomic Manipulation and Antimicrobial Resistance

Response: The present manuscript was not submitted under the scope of the special issue CRISPR-Cas in Genomic Manipulation and Antimicrobial Resistance.

I detected some mistakes:

  1. sometimes M. tuberculosis is not in italic style.

Response: Corrected.

  1. line 100 and 101 missing μ

Response: Corrected.

  1. línea 144 Figura complementaria 1.

Response: The figure cited in the text is indeed Figure S1, not Figure 3.

  1. All the references must be checked; they are not in accordance with the style of journal.

Response: Done, all references were checked and corrected.

Round 2

Reviewer 3 Report

The authors improved the manuscript with the reviewers' comments. I suggest the paper can be accepted.